# A Conceptual Framework for Modeling Social Risk Tolerance for PPP Projects: An Empirical Case of China

**Weiyan Jiang [1,\*], Jingshu Lei [2], Meiyue Sang [2], Yinghui Wang [2] and Kunhui Ye [2]**

1  Business School, Southwest University of Political Science and Law, Np. 301, Baosheng Avenue, Yubei District, Chongqing 401120, China
2  School of Management Science and Real Estate, Chongqing University, Chongqing 400045, China; jincqliu@shinyway.com.cn (J.L.); meiyuesang@cqu.edu.cn (M.S.); wangyinghui13@163.com (Y.W.); Kunhui_YE@cqu.edu.cn (K.Y.)
\*  Correspondence: jiangweiyan@swupl.edu.cn

**Abstract:** Public–private partnerships (PPPs) are a useful approach that allows the public sector to collaborate with private investors in financing, implementing, and operating public sector facilities. Over the past few decades, the occurrence of social risks and the vulnerability of PPP projects to these risks have caused numerous project failures. While practitioners claim to manage the social risks of PPP projects, little effort has been made to explore the proper ways of doing this. In this study, we present a social risk tolerance (SRT) concept and propose a model to quantify the tolerance of PPP projects to social risks. One hundred and twenty-three PPP projects were collected from China for model validation. The results indicate a positive relationship between SRT values and project size and that the SRT has diminishing marginal values. This paper presents a new concept in PPP research and provides an appropriate approach for managing the social risks of PPP projects. The research findings can help both the public and private sectors understand the social risks associated with PPP projects and determine effective countermeasures to control these risks.

**Keywords:** PPP; social risks; social risk tolerance; sustainable construction; China

## 1. Introduction

Public–private partnerships (PPPs) originated in western countries [1] and have been applied worldwide to procure infrastructure facilities and public services. Evidence has demonstrated that PPPs have multidimensional advantages, including mobilizing social capital, relieving financial burdens on the government, improving project delivery efficiency, and generating social welfare programs [2,3]. According to statistical data from the World Bank [3], global private investment in PPP-related infrastructure projects had amounted to USD 1788 billion by 2018. The United Kingdom, Canada, and Australia have established public procurement departments to deal with the enormous demand for PPPs. In China, there are 14,220 projects included in the national PPP project management database, with a total investment of USD 2.75 trillion that involve 19 industries, including transportation and municipal administration.

PPP approaches facilitate the development of public facilities and creating a public service system. Finding sufficient social capital from the private sector is a prerequisite for PPP undertakings. However, private investors usually pay less attention to non-profit businesses and related issues such as societal interests and project-related social risks. Some scholars have pointed out that PPP projects should satisfy the demands of governments for infrastructure services and should also address social welfare issues, including land expropriation and environmental degradation [4]. According to Marques and Berg (2011), contractual parties involved in PPP must identify and assign risks for effective mitigation [5]. One of the primary reasons for this is that social risks can translate into construction challenges and can incur unexpected costs for private investors [6].

A variety of social risk themes have been examined in current research conducted in the construction area. For example, Kemp et al. (2016) identified the social risks of global mining [7]. They disclosed that the risks are likely to a have massive impact on the construction projects achieving its goals. Otieno and Loosen (2016) examined a few centralized solar power generation projects in California and discovered the frequent occurrence of social risks in PPP projects [8]. In referring to the Sydney Intercity Tunnel, Johnston and Gudergan (2016) deemed that unforeseen political and social risks were the root cause of PPP failure [9]. Such prominence has inspired researchers to call for more resources, time, and attention to improve social risk management [5,10]. This essentially requires a two-faceted approach: assessing social risk loss and the probability of risk occurrence in the early construction phase of the project. Although such an assessment can help us to understand whether a PPP project can resist and recover from social risks, tools for diagnosing these social risks are very limited. In effect, PPP projects are vulnerable to social risks, suggesting that a precise social risk diagnosis deserves much attention.

In adhering to the principle of social risk management, the World Bank Group (2009) devised a set of indicators that included public participation, employment rate, and information disclosure [11]. They also built a social risk assessment technology system in order to complement qualitative inference and quantitative analysis. Simon (2012) compared the one- or two-dimensional Monte Carlo method, micro exposure event analysis, and probabilistic risk assessment methods using data from the Gemosanwes Superfund website [12]. In light of the Bayesian hybrid risk assessment theory, Ung (2018) proposed transforming expert qualitative judgments into probabilistic risk results for offshore engineering systems [13]. Cheng and Lu (2015) established a risk assessment model by combining fuzzy reasoning, failure mode, and effect analysis and discussed the severity and criticality of risk events [4]. Finally, Furuncu and Sogukpinar (2015) established a cloud computing service risk assessment model and used algorithms to create security solutions for engineering projects, including those involving infrastructure [14]. The richness and availability of risk assessment tools in the relevant literature suggest that developing an approach to quantify the social risks involved in PPP projects has been supportive and feasible.

Recent years have witnessed the prosperity of PPP projects in China [1], providing a useful resource for examining the social risks in developing countries. This study aims to present a concept of social risk tolerance (SRT) for PPP projects and proposes a model to quantify the ability of PPP projects to withstand social risks. Managing social risks requires the process of identifying, analyzing, prioritizing, treating, and monitoring risks to be conducted properly [15–17]. Furthermore, developing solutions is necessary to exert control, provide early warning, and compensate for social risks [18]. Therefore, the current research can favor both the public and private sectors, stressing the importance and value of social risks and shedding some light on mitigating social risks in PPP project delivery.

## 2. Literature Review

### 2.1. Social Risks in PPP Projects

Social risk, a term originating from the "risk society theory" (Beck and Ritter, 1992), refers to the possibility of causing social conflicts, endangering social stability, and disturbing social order. Social risk is a two-dimensional concept presenting the possibility of social crisis and the degree of conflicts caused by that social crisis [2]. The literature on social risk has been focused on a few important topics, such as social exclusion and education, in the field of sustainable development [19]. According to Rucinska (2015), social risks can disrupt the quality of life and the sustainable growth of economies [20], suggesting that social risks be managed effectively. A range of indicators, including the breadth, intensity, persistence, and magnitude of social conflicts, have thus recommended the measurement of social risks [1,7]. Furthermore, researchers have advocated a holistic approach for assessing social risks to underpin relevant decision-making [21,22].

In the construction context, social risks are compounded by the interaction of the client, contractors, consultants, and suppliers [23]. Scholars have highlighted that the social

risks that are related to construction projects extend to the area of environmental pollution, land acquisition, the demolition of existing buildings, and these projects sometimes being unsafe [24]. These risk factors are economic, environmental, safety, and societal [17]. If proper measures are not taken, social conflicts will be converted into social risks (such as petitions, processions, demonstrations, crimes) [10,25]. Regarding PPP projects, the information asymmetry between public sector managers and private sector investors has become an obstacle to the close cooperation between the two sides, which is deemed a cause of social risks [26]. To mitigate information asymmetry, the public and private sectors claim to jointly manage social risks to guarantee the successful delivery of PPP projects [27].

### 2.2. Definition of SRT

Risk tolerance, a proxy for the level of risk that an organization is willing to accept, has well been examined in investment decision-making science [28]. This term is concerned about the degree of risk that an investor feels comfortable with or the extent to which an investor can handle. In biology, a biological function acts at or near the optimum point, weakens when it tends to both ends (maximum and minimum), and then is inhibited, which is called the "law of tolerance" [29]. Likewise, social risk tolerance (SRT) is coined to present the amount of loss caused by social events that the main stakeholders are able to tolerate or how much risk they would like to face.

Social risks are related to industrial sectors and human behaviors, implying that SRT contains a behavioral preference. Some scholars have stated that social risks usually occur in the financial [30], cultural [31], medical [32], and manufacturing sectors and that the definition of relevant concepts should account for the differences in industrial backgrounds. Previous studies have also ascribed social risks to the interactions between teenagers [6], per capita income [33], and citizen crime [34], indicating that SRT should be as specific as possible. In essence, social risks play a crucial role in impacting the performance of construction projects. The impacts are direct, indirect, or mediated through behavioral risk factors. For this reason, a social stability risk assessment system has been established by China's Sichuan Government to draw societal attention to the social risks of large-scale construction projects. This risk assessment system is widely recognized, as it may inform the extent to which an entity tolerates social risks.

PPP projects are characterized by large investments, long cycles, complex contracts, and arduous coordination tasks. The development of PPP projects is subject to some problems, such as interest constraints and information asymmetry. A PPP project has an interlocking relationship between stakeholders. The greater the social risks, the larger the constraints imposed by project stakeholders. The demands and interests of PPP stakeholders fluctuate from one region to another, leading to uncertain factors and social contradictions along the construction process. Social risks are also associated with extreme individual or group events [13]. Therefore, the effective management of social risks during PPP projects is not an easy task.

### 2.3. SRT Measurement for PPP Projects

The tolerance that PPP projects have to social risks is dependent upon the project type. Tolerance to one social issue may be very broad, but this tolerance may diminish quickly, implying that the tolerance of an organism to the same ecological factor is situation-based. Moreover, biological tolerance varies with age, season, and habitat, describing the dominant role of biological heterogeneity in biological tolerance. Similarly, the evaluation of SRT of PPP projects should be built on the proper selection of both scientific and reasonable methods.

It is necessary to link the SRT of PPP projects to the heterogeneity of PPP projects. Wu et al. (2008) examined the on-site detection and monitoring data of large hydro-power project construction, identified the internal risk factors, proposed basic risk assessment criteria, constructed a three-level risk assessment system, and monitored the dam's operation risk [35]. In noting large-scale infrastructure construction projects such as the South-to-

North Water Diversion Project and urban subway projects, the entropy weight coefficient, the analytic hierarchy process, and other methods are often employed to explore the risk management system that is applicable to the decision-making stage, the implementation stage, and the operation stage of construction projects [4,16,36,37].

### 3. Model Development

Based on the above discussion, we proposed an SRT measure for SRT projects, for which the project's uniqueness and SRT definitions are addressed. Considering that there are no directly available quantification models, we synthesized the existing risk assessment models with risk factors and the project scale described below.

#### 3.1. Calculating Risk Indexes

A generic risk index was adopted to measure the numerical value of a PPP project's social risks by being given a set of risk factors. Assuming that there are $n$ risk factors for $m$ projects, the occurrence probability $(p_{ij})$ of risk factor $j$, and the impact of risk $(q_{ij})$ on project $i$, the risk index $(R_{ij})$ of risk factor $j$ for project $i$ can be calculated.

$$R_{ij} = p_{ij} \times q_{ij} \ (1 \leq i \leq m, 1 \leq j \leq n) \tag{1}$$

We took a further step to calculate risk factor $j$'s weight $\omega_j$. The methods available include expert scoring and entropy, depending on data availability. Thus, factor $i$'s risk index can be calculated as follows:

$$a_{ij=}R_{ij} \times \omega_j \tag{2}$$

#### 3.2. Introducing a Scaling Coefficient

Project scale is treated as the main parameter of a construction project's risk tolerance. In appreciating that a PPP project's size might be massive but that its social risk index is relatively small, we introduced a scaling coefficient $C_i$ to convert project investment into a comparatively small value for the convenience of comparison. We can assume that project investment size $b_i$ contains intervals, including the maximum and minimum value of the project investment amount, signifying $z_{min \cdot k}$, $z_{max \cdot k}$, respectively. Such transformation fits with public perceptions on the project scale. On most occasions, public views on project size reflect a few major types: mega-projects, large projects, medium projects, and small projects. In addition, project size is often based on the total financial resources that are available, the complexity of deliverables to be constructed, timeframe requirement, and the team members involved. Therefore, a scaling coefficient can be calculated as follows:

$$C_i = \begin{cases} k + \frac{b_i - z_{min \cdot k}}{z_{max \cdot k} - z_{min \cdot k}} \ (z_{min \cdot k} \leq b_i < z_{max \cdot k}, k = 1, 2, 3, \ldots, g-1, i \in (1, m)) \\ g \ (z_{min \cdot k} \leq b_i, k = g, i \in (1, m)) \end{cases} \tag{3}$$

#### 3.3. Calculating Risk Factor-Based SRT

As defined above, the SRT refers to the potential of resisting social risks that have been rendered by a set of risk factors. Thus, a PPP project's tolerance to social risks results from its tolerance to social risk factor $i$, which is called $s_{ij}$. Supposed that a risk index is zero, the project can completely resist such risk factors. Thus, the calculation formula assumes "risk tolerance = 1 − risk index/project size". Such calculation ascertains no social risks to be accounted for, and the project's social risk tolerance should be full, assigning a value of one. By substituting Equations (2) and (3) into Equation (4), a PPP project's SRT to risk factors $(s_{ij})_{m*n}$ can be considered.

$$s_{ij} = 1 - \frac{a_{ij}}{c_i} \tag{4}$$

*3.4. Calculating SRT Values*

Weightings were introduced to accumulate a PPP project's total SRT value from its tolerance to a single risk factor. Entropy weighting is an effective method to determine objective weight based on actual values. Moreover, this method can avoid the influence of subjectivity factors and can normalize the original data to meet monotonicity requirements, scale independence, and total amount constancy. Hence, we employed entropy weighting to derive the project's SRT coefficient based on a single factor risk tolerance value.

The SRT matrix $S = \left( s_{ij} \right)_{m*n}$ can be standardized using Equation (5). The information entropy $E_j$ for risk factor $j$ is calculated using Equation (6), where $p_{ij} = \frac{y_{ij}}{\sum_{i=1}^{m} y_{ij}}$. If $p_{ij} = 0$, let $\lim_{p_{ij} \to 0} p_{ij} ln p_{ij} = 0$.

$$y_{ij} = \frac{s_{ij} - \min_{1 \leq j \leq n}(s_j)}{\max_{1 \leq j \leq n}(s_j) - \min_{1 \leq j \leq n}(s_j)} \tag{5}$$

$$E_j = -\ln(m)^{-1} \sum_{i=1}^{m} p_{ij} ln p_{ij} \tag{6}$$

The information entropies for all of the risk factors are $E_1, E_2, E_3, \ldots, E_n$. Based on these coefficients, the entropy weight $W_j$ for risk factor $j$ can be represented as:

$$W_j = \frac{1 - E_j}{\sum_{j=1}^{n} \left( 1 - E_j \right)} \tag{7}$$

Equation (8) was developed for project $i$'s SRT coefficient $(Z_i)$:

$$Z_i = \sum_{j}^{n} s_{ij} W_j$$
$$= \sum_{j}^{n} (1 - \frac{\sqrt{a_{ij}}}{c_i})(\frac{1 - E_j}{\sum_{j=1}^{n} \left( 1 - E_j \right)}) \tag{8}$$

## 4. Data Collection and Analysis

*4.1. The Sample*

In 2014, the Ministry of Finance of China established a public–private partnerships center, launched a national PPP project management database, and regularly publicized PPP project reports. Making this information available facilitates the examination of the SRT of PPP projects. We collected 123 PPP project feasibility study reports. The sample projects were distributed over seventeen provinces, including Guangdong (15), Fujian (21), Sichuan (16), Hebei (13), and Yunnan (10). They comprised 52 transport projects and 71 municipal projects.

*4.2. Risk Indexes*

Included in the feasibility reports are social risk assessment results provided by a third party, including risk probability $p$, potential impacts $q$, risk degree $R$, and risk index $a$ for all risk factors. These indices were released in the national PPP project management database, suggesting that the data have good credibility that has been recognized by the Ministry of Finance of China. In addition, there are 49 social risk factors that have been formulated by the National Development and Reform Commission. However, some of the factors are defined obscurely and have been removed. Thus, we retained a social risk framework composed of 32 risk factors, as shown in Appendix A.

The basic information and total investment $b_i$ per PPP project were counted, and the social risk $a_{ij}$ index matrix $A = (a_{ij})_{m*n} (1 \leq i \leq m, 1 \leq j \leq n, m = 123, n = 32)$ can be calculated based on Equation (2). The risk index results span from the maximum value (0.12) to the minimum value (0). The average value is 0.006, the mode value is 0, and the median value is 0.

### 4.3. Scaling Coefficients

All of the cited PPP projects were divided into five groups (Table 1), of which $b_i$ is project $i$'s investment. The derived scaling coefficients are 5 (maximum), 0 (minimum), 2.787 (mean), and 2.703 (median).

**Table 1.** Five levels of scaling coefficients.

| Level | Investment [a] | Number of Projects (PCS) | Scaling Coefficient $C_i$ |
|---|---|---|---|
| I | (0, 500) | 33 | $1 + \frac{b_i - 0}{5 - 0}$ |
| II | (500, 1000) | 32 | $2 + \frac{b_i - 5}{10 - 5}$ |
| III | (1000, 5000) | 40 | $3 + \frac{b_i - 10}{50 - 10}$ |
| IV | (5000, 10,000) | 10 | $4 + \frac{b_i - 50}{100 - 50}$ |
| V | (10,000, +∞) | 8 | 5 |

[a]-million Yuan (CHN).

### 4.4. Calculating SRT Values

The calculated risk factor indexes fall into the range of (0.01, 0.1) and demonstrate insignificant numerical fluctuation. To increase the dispersion and difference of the risk factor index, we calculated the SRT matrix $S = (s_{ij})_{m*n} (1 \leq i \leq m, 1 \leq j \leq n, m = 123, n = 32)$ based on Equation (4). According to Equation (8), the minimum value of SRT $Z_i (1 \leq i \leq 123)$ is 0.9259, the maximum value is 0.9950, and the average value is 0.9739 (see Figure 1). The number of items scattered in the range of (0.9778, 0.9864) was the largest, with 35 in total; the number of items scattered in the range of (0.9691, 0.9778) was the second.

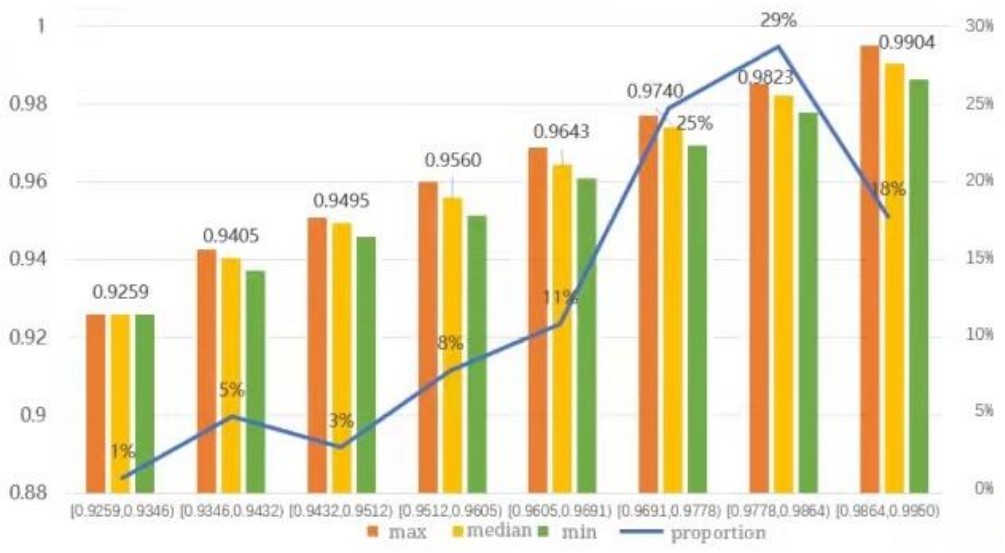

**Figure 1.** Distribution of SRT values.

*4.5. Verification*

In engineering economics, net present value (NPV) reflects the difference between the present value of future cash inflow and future cash outflow. It functions as a proxy for project size. Thus, we used NPV as the dependent variable and SRT coefficients as the independent variable to test the fit of the model results with the general sense. A rule of thumb is that given a fixed social risk, the larger the project size, and the greater the tolerance to the social risk. Hence, the SRT model can be feasible and effective if the results align with such a rule. To reduce the absolute value of NPV, ln (NPV) is considered. Taking SRT value as an independent variable $x$ and ln (NPV) as a dependent variable y, the adjusted coefficient $R^2$ is 0.618, and the regression equation was $y = 119.742x - 106.495$. As indicated by the residual analysis, the value of the Durbin Watson test was 1.801, which is close to 2, suggesting that the derived SRT data were completely independent (Figure 2). Moreover, the scatter points of the P-P diagram are similar to straight lines. Therefore, the residuals obey normal distribution and meet the modeling requirements. The higher the project size, the higher the SRT, suggesting that the model results fit public perception.

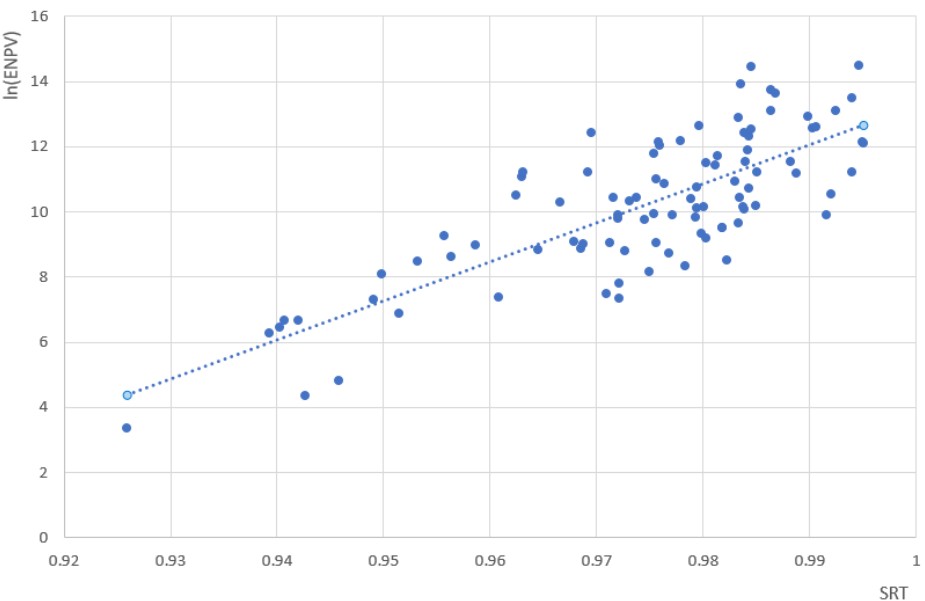

**Figure 2.** Regression results of SRT coefficients and NPV.

## 5. Model Application

The risk factors of the SRT model are multiple and flexible, posing a strict requirement on the quality of the original data. Data for calculating the SRT model are often subject to great difficulties in terms of data collection. However, considering that the 123 PPP projects collected in this study were sufficient for further modeling, we utilized the calculation results of the SRT values to provide a more intuitive and convenient method to satisfy the fast decision-making requirements in reality. Taking project size as the independent variable $u$ and SRT as the response variable $v$, the resulting logarithmic model was $v = 0.0082 \ln u + 0.8794$, as shown in Figure 3. The Durbin Watson test value (1.824), zpred-zresid scatter plot, residual histogram, and P-P plot of the residual analysis results show that the residual variable obeys normal distribution and meets the modeling requirements. The curve results demonstrate that the SRT curve sustains significant waving among those projects in small sizes, and the curve tends to be flat and subsequently slows down with an increase in project size. A marginal analysis curve is also drawn to depict the rules of marginal SRT. The SRT coefficients of SRT projects have a marginal decreasing tendency. Alternatively, the increment of risk tolerance obtained by each continuously increasing investment unit decreases with an increase in project size.

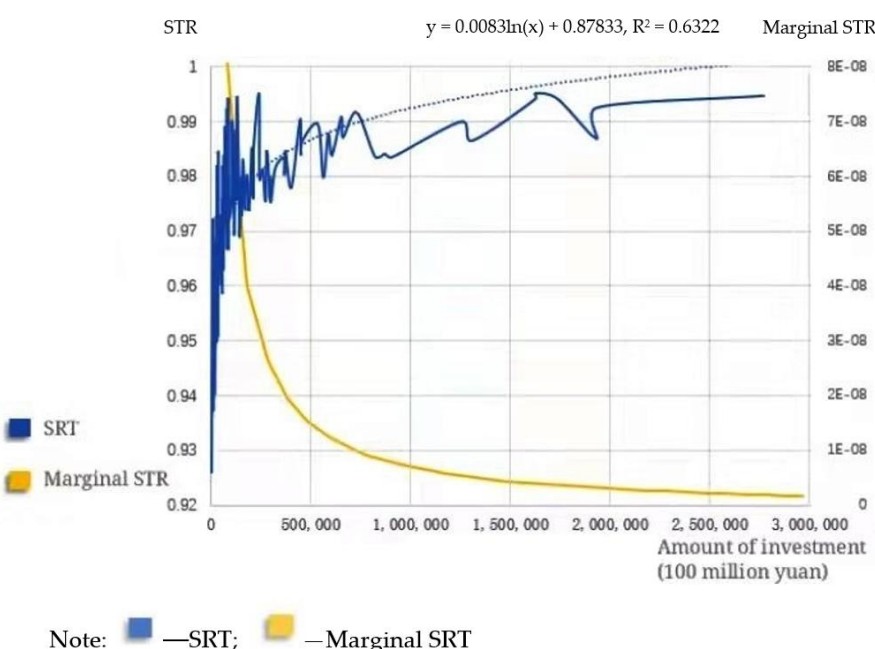

**Figure 3.** SRT and marginal SRT curves.

## 6. Findings and Discussion

### 6.1. PPP Projects' Tolerance to Social Risks

PPP modes have been extensively adopted to supply high-quality services and public goods, relieve fiscal burdens, and promote urbanization. However, the involvement of private capital in the development of public facilities favors the emergence of social risks in PPP projects. PPP model failures may be encountered if the mitigation strategies for social risks are ineffective. Based on the theory of biological tolerance to the environment, this study presents the term "social risk tolerance" (SRT) to illustrate the ability of a PPP project to resist social risks. The concept of SRT is vital to the success of PPP project management, suggesting that stakeholders ought to strike the trade-off between risk and reward and should adopt an integrated risk management framework often. It was found that a PPP project's SRT is three-dimensional, including the probability of occurrence, the magnitude of impacts, and project size. PPP projects in the public and private sectors are particularly sensitive to social risks, and the expectation is that PPP projects are developed in a socially responsible manner. Therefore, establishing an effective risk-sharing mechanism between the public and private sectors is prominent in mitigating social risks.

The SRT model further develops traditional risk assessment methods. Previous research on construction project risk management is concerned with identifying and controlling risk factors [2,7,12]. For example, scholars have highlighted construction-related (e.g., external suppliers) (Sarker et al. 2016) and systematic risk factors (e.g., inflation, recessions, and natural disasters) [38]. It has been advocated that although social risks may not be eliminated, PPP projects must mitigate and control social risks properly. Therefore, they require due diligence in the early decision-making stages of the project. However, a PPP project is vulnerable to social risks that can impact society and cause social unrest and turbulence. In the PPP area, social risks can be triggered from conflicts of interest among different stakeholders and have been a leading disruptive force PPP project management improvement.

### 6.2. The Relationship between SRT and Project Size

The tolerance of a PPP project to social risk is determined by a combination of factors, such as project goals, project team experience, and project complexity. It has been found that when the project size increases, PPP projects will follow an SRT development trajectory.

From the perspective of project affordability, the larger the PPP project size, the greater the affordability, and as such, more attention should be paid to social risks.

Such inconsistent relationships between the SRT and project size can be justified from four aspects. First, both the public and private sectors stress the importance of project construction schemes in PPP modes. Their joint efforts are usually made to predict the uncertainty of risks and to negotiate a reasonable distribution of interests between stakeholders. The more dimensions embraced in the scheme, the higher the public's reliance in searching for large-scale social capitals to transfer social risks onto. Second, when the PPP project is larger, the public sector has to face longer timeframes and higher costs associate with the project, which reduces the occupation and impact of public resources and the control social risk factors such as environmental pollution. Third, an increase in a PPP project's size indicates more penetration of the PPP project into the social lives of the public. Solving mass disputes in a timely manner and reducing conflicts of interest are viewed as key success factors for the project. Finally, PPP projects with large investment scales are more likely to attract the attention of the media and the public, calling for an increase in the tolerance of a project to social risks.

The present study complements previous studies demonstrating the significant relationship between SRT and project size. Risk tolerance measurement might not be precise. One of the reasons for this is that measuring both an investor's emotional and financial ability in withstanding losses is subjective. Thus, the measurement has to account for net worth, income, knowledge, sophistication, and proximity to retirement as risk tolerance factors. In PPP projects, most social risks are intangible and invisible at early stages of the construction projects and are based on how many risks the public and private sectors can tolerate. These risks are based on what the public and private sector deem to be acceptable to them and may be conservative, moderate, or aggressive. Knowing the vital role of project size in framing social risk tolerance helps the public and private sectors determine an effective approach to managing these risks in the coming construction phase.

*6.3. PPP Projects' Marginal SRT*

The marginal analysis method of economics is introduced to reveal the relationship between project size and the SRT. It was found that the SRT value of PPP projects has a marginal diminishing tendency. This phenomenon is real, as explained from two perspectives. First, PPP project investors can control social risks by increasing investment and by formulating risk aversion measures. However, the project's tolerance to social risks is limited, as it does not need endless investment to improve its resistance to social risks. Second, the investment diversity determines that investors will allocate resources that are in line with investment plans. When the project size is small, the SRT has a high level of sensitivity to change. The primary condition to ensure the smooth progress of the project is to reconcile the contradictions between the public and private sectors.

A better understanding of the marginal tolerance of PPP projects supports project management teams in gauging and embarking on the delivery of successful projects. The inclusion of marginal tendency in measuring the SRT is an extension of previous studies. A project's tolerance to social risks is not substantial if the project size is small. At this moment, small emergencies may lead to project failure. On the other hand, a larger project requires a better ability to handle the technical complexity and toughness of project tasks. Thus, the SRT of PPP projects becomes stronger in resisting the impulse of social risks when the size of the project increases. However, the SRT sustains relatively stability when project size exceeds a certain level, depending on the characteristics of the project. The reasons for such a changing trajectory might be that the larger the size of a PPP project, the stronger the social risks the project will face. Therefore, private investment will be made conservatively if the PPP project's social risk tolerance is low.

## 7. Conclusions

The contradiction between the public welfare of PPP projects and the profit pursuits of social capital is often transformed into social risks. Therefore, the success of PPP projects depends on the effective manipulation of social risks. This paper presents the concept of SRT for PPP projects, establishes an evaluation model for SRT, and links the SRT of PPP projects with project size. It was found that given a project's size, a PPP project's SRT is computable, providing rapid decision-making support for both the public and private sectors. The study provides a new approach for determining a reasonable amount of investment and adjusting the project structure, which will help construction project teams understand the social risks of PPP projects. Furthermore, risk decision-making and risk prevention can map out data to establish social risks as early warning mechanisms of PPP projects.

While the study presents new research on the social risks of PPP projects, some shortcomings were evaluated in this study. First, social risk is an abstract concept, and there are differences among different countries or regions. Second, there are many kinds of PPP models, and the project risks might be inconsistent. Third, the SRT model for construction projects will vary from one country to another, and the relationship between SRT and project scales is open for revision.

**Author Contributions:** W.J.: conceptualization, methodology, analysis, writing—original draft, writing—review and editing, project administration; J.L.: conceptualization, analysis, writing—review and editing; M.S.: analysis, writing—review and editing; Y.W.: conceptualization, data collection, methodology; K.Y.: data collection, methodology, visualization. All authors have read and agreed to the published version of the manuscript.

**Funding:** This research was funded by the Humanities and Social Science Foundation of the Ministry of Education of China (Grant Number 19YJC630065).

**Institutional Review Board Statement:** Not applicable.

**Informed Consent Statement:** Not applicable.

**Data Availability Statement:** Data are available from the authors upon request.

**Conflicts of Interest:** The authors declare no conflict of interest.

## Appendix A. List of Social Risk Factors

| Sequence Number | Risk Factor | Explanation |
|---|---|---|
| V1 | The risk of the legitimacy and rationality of the project being questioned | Whether it conflicts with existing policies, laws, and regulations, and whether it has sufficient policy and legal basis; Whether it has been demonstrated by rigorous scientific feasibility studies |
| V2 | The risk of discomfort to the changing living environment | During the construction period, the public were disturbed by the outside world to some extent, which caused the public' uneasiness and worry |
| V3 | Risk of social conflict arising from the project (influence of stakeholders) | Project stakeholders have conflicts of interest, which will interfere with project progress and affect normal production and safety |
| V4 | The legality and compliance of project approval procedures | Whether the project approval complies with relevant national laws and regulations; whether to adhere to the strict examination and approval and approval procedures; |
| V5 | Compliance with industrial policy and development plan/project feasibility | Whether it conforms to the requirements of industrial policy, overall plan, special plan; and industry access; and whether it conforms to the regional planning and development status |
| V6 | Public participation in project approval | Whether opinions are widely heard and whether public opinions can be given truthfully and can receive timely feedback |

| V7 | Land and housing expropriation and requisition scope | Whether the construction land conforms to the overall requirements of adjusting measures to local conditions and economizing the use of land for self-use, the relationship between the scope of housing expropriation and demolition and the demand for engineering land, and the local land use planning, etc. |
|----|----|----|
| V8 | Compensation funds for expropriation and requisition of land and houses | Source of funds, quantity, implementation situation |
| V9 | The employment and life of the land expropriated farmers/the risk of the people's worry about the livelihood security/the social risk of immigration | Mass society, health care programs and their implementation, skills training, and employment plans, etc. |
| V10 | Resettlement housing quantity and quality | Total housing supply ratio, regional housing supply ratio of the current year, forward housing/existing housing ratio, housing supply status and planning supporting level, transportation and surrounding living supporting facilities, integration degree of resettlement residents and local residents |
| V11 | Compensation standards for land and housing expropriation and demolition | The relationship between the physical and monetary compensation and the market price and the relationship between the compensation standard of similar land recently |
| V12 | Procedures and plans for compensation for expropriation and demolition of land and housing | Whether to carry out land and house expropriation compensation work in accordance with the procedures prescribed by national and local laws and regulations, whether to solicit public opinions on the compensation plan, etc. |
| V13 | Other compensation to the place | Compensation scheme for construction damage to buildings and compensation scheme for people affected by various living environments due to the implementation of the project |
| V14 | Project scheme | Generally, risk factors of engineering safety and environmental impact occur at the same time, which can be analyzed according to specific projects (for example, inflammable, and explosive projects should consider the possible damage within and outside the safety distance; the safety and environmental protection standards implemented in the technical scheme are low and inconsistent with the acceptance capacity of the public.) |
| V15 | Financing and security | The feasibility of fundraising scheme, whether the fund guarantee measure is sufficient |
| V16 | Emission of air pollutants | Whether to strengthen the construction stage management measures, whether to take dust removal measures for sand and gravel, strengthen road maintenance and cleaning work, whether to reduce the working time in windy weather |
| V17 | Emission of water pollutants | Whether the wastewater generated by the project is purified and treated, whether there is an emergency plan for water pollution accidents, and whether the channels for local residents to report and appeal are kept open |
| V18 | Noise and vibration effects | Whether the construction sequence and construction time are reasonably arranged, whether the equipment with large vibration is installed with shock absorption facilities, whether the construction facilities are reasonably arranged and the construction management is strengthened |
| V19 | Solid waste and its secondary pollution (garbage odor, leach, etc.) | Whether solid waste can be included in the sanitation collection and transportation system to ensure daily clearance; construction waste, bulky waste, engineering muck, toxic and harmful solid waste such as medical waste can be handled by qualified collection and transportation units and so on |
| V20 | Public open activity space, green space, water system, ecological environment, and landscape | The change of public activity space and quality, the surface of public green geology and quantity, the change of water system, ecological environment, and community landscape |

| V21 | Soil erosion/geological disaster/soil and water conservation | Possible changes of terrain, vegetation and soil structure, possible effects of waste soil and residue, whether there is a soil and water conservation plan, etc. |
|---|---|---|
| V22 | The removal and disposal of soil | Whether the removal and disposal of soil meet the requirements of environmental protection |
| V23 | Project "five system" construction | Legal person responsibility system, capital fund system, bidding system, establishment system and contract management |
| V24 | Project unit six management system | Approval or approval management, design management, budget management, construction management, contract management, labor management, etc. |
| V25 | Construction plan (risk caused by poor construction and traffic organization plan) | The connection between the construction measures and the construction time sequence of adjacent projects, the relationship between the implementation process and sensitive time points (two sessions/college entrance examination), whether the construction cycle arrangement interferes with the production and life of surrounding residents, etc. |
| V26 | Civilized construction and quality and safety management | Violate the relevant regulations of civilized construction and quality and safety management, resulting in environmental pollution, water cuts, power cuts, gas cuts, traffic impacts, and other emergencies and quality and safety accidents |
| V27 | Social stability risk management system | Whether the project unit of social stability risk management system and the local government have fully communicated with each other about the project, whether they fully understand the risks of social stability and do their respective duties, whether they have established the responsibility system and linkage mechanism of social stability risk management, whether they have formulated the corresponding emergency response plan, etc. |
| V28 | Influence on surrounding traffic/poor traffic organization plan during construction | The construction plan should consider the travel and traffic of the surrounding people (temporary sidewalk setting, temporary parking lot site arrangement, temporary bus stop layout, etc.), the change of the bus traffic situation around the project during the operation period, the matching degree of the increased traffic flow of the project with the surrounding road network, the impact of the entrance and exit setting on the surrounding people, etc. |
| V29 | Construction safety, hygiene and Occupational health/risk of public concern for project safety | Management of soil-moving vehicles and other transport vehicles, hazards, harmful factors and safety management systems, health and occupational health management, emergency handling mechanism, etc. |
| V30 | Social security and public safety | Construction team size, management mode, user analysis during operation (user source, quantity, mobility, cultural quality, age distribution, etc.) |
| V31 | Media public opinion orientation and its influence | Whether to obtain media support, whether to coordinate and arrange authoritative and credible media to publicize project construction information and give positive guidance, whether to receive media attention and public opinion-oriented information |
| V32 | Workers' wages/labor employment | Whether the labor and employment in the construction process is standard, whether the system is perfect, whether to protect the rights and interests of workers |

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
