# Peer review of "A Conceptual Framework for Modeling Social Risk Tolerance for PPP Projects: An Empirical Case of China"

_buildings, doi:10.3390/buildings11110531_

Round 1
Reviewer 1 Report
The article addresses the social risk issue in PPP contracts by developing an empirical case study involving a set of PPP projects in China. The topic is interesting and the article is well written. Although the paper has some merit, there are some issues that should be clarified by the authors.
The major issue is that the authors focus on PPP projects but what is the difference relative to the public work contracts? The concept of risk is quite important in PPP contracts but this is not approached here and it should be. For example, see the concept of risk of ISO 39000 or some seminal papers about it (see, for example: Marques and Berg (2011). Journal of construction engineering and management 137 (11), 925-932). Even in PPP contracts social risk is completely different depending on the commercial risk, if there is no direct relationship with the customers (they do not pay tariffs/tolls, …) social risk is the same in PPP contracts as public work contracts. Also, why China? The authors should justify the reasons to choose China and the PPP particularities in this country (see, for example the recent paper of Pu et al. (2021). A bibliometric and meta-analysis of studies on public–private partnership in China. Construction Management and Economics.
Author Response
The major issue is that the authors focus on PPP projects but what is the difference relative to the public work contracts?
Response: Thank you for the comments. Please find the revision on line 57.
“relative to public work projects, a PPP project is more vulnerable to the impulse of social risks due to the involvement of private investment. This gives the suggestion that a precise social risk diagnosis should be given to PPP projects at an earlier stage.”
The concept of risk is quite important in PPP contracts but this is not approached here and it should be. For example, see the concept of risk of ISO 39000 or some seminal papers about it (see, for example: Marques and Berg (2011). Journal of construction engineering and management 137 (11), 925-932). Even in PPP contracts social risk is completely different depending on the commercial risk, if there is no direct relationship with the customers (they do not pay tariffs/tolls, …) social risk is the same in PPP contracts as public work contracts.
Response: Thank you for the comments. Please find the revision on lines 42-45.
“According to Margues and Berg (2011), risk is reflected and assigned to PPP contractual parties, calling for effective mitigation. One of the primary reasons is that social risks can transform into construction costs and incur unexpected costs for private investors if they are not handled appropriately.”
Also, why China? The authors should justify the reasons to choose China and the PPP particularities in this country (see, for example the recent paper of Pu et al. (2021). A bibliometric and meta-analysis of studies on public–private partnership in China. Construction Management and Economics.
Response: Thank you for the comments. Please find the revision on lines 82-83.
“Recent years have witnessed the popularity of PPP projects in China, providing a breeding ground for examining the measurement of social risks in other developing countries.”
Meanwhile, the conclusion section has been improved as expected.
Reviewer 2 Report
I attach a copy of the paper (marked up).
The article subject has merit.
The authors need to address, why is the proposed quant method of value? What does this study provide that's unique or unknown?
The paper requires significant additional work as indicated in the mark-up. Most importantly, the authors assert terms/concepts without sufficient definition.
Graphics need to be expanded, otherwise the type is too small.
The article may have merit, at this point it's difficult for this reader to determine the merits.
Research keywords -- capitalization is inconsistent. I recommend deletion of e-gov't.

Author Response
Thank you so much for the detailed comments. We would like to accept all of the comments and have made major changes to the paper as requested. Kindly please check the revised paper to find our revision. Should we clarify, please feel free to contact us.
Round 2
Reviewer 1 Report
The authors improve substantially the paper and I recommend its publication.